# Fungal Plasma Membrane H^+^-ATPase: Structure, Mechanism, and Drug Discovery

**DOI:** 10.3390/jof10040273

**Published:** 2024-04-08

**Authors:** Chao-Ran Zhao, Zi-Long You, Lin Bai

**Affiliations:** 1Department of Otolaryngology Head and Neck Surgery, Beijing Tongren Hospital, Capital Medical University, Beijing 100730, China; 2Beijing Key Laboratory of Nasal Diseases, Beijing Institute of Otolaryngology, Beijing 100005, China; 3Department of Biophysics, School of Basic Medical Sciences, Peking University, Beijing 100083, China

**Keywords:** fungi, plasma membrane H^+^-ATPase, structure, drug target

## Abstract

The fungal plasma membrane H^+^-ATPase (Pma1) pumps protons out of the cell to maintain the transmembrane electrochemical gradient and membrane potential. As an essential P-type ATPase uniquely found in fungi and plants, Pma1 is an attractive antifungal drug target. Two recent Cryo-EM studies on Pma1 have revealed its hexameric architecture, autoinhibitory and activation mechanisms, and proton transport mechanism. These structures provide new perspectives for the development of antifungal drugs targeting Pma1. In this article, we review the history of Pma1 structure determination, the latest structural insights into Pma1, and drug discoveries targeting Pma1.

## 1. Introduction

Due to the limited types of antifungal drugs available and the emergence of drug-resistant strains, fungal diseases are a growing threat to human health. Approximately 1.7 billion people have skin, nail, or hair fungal infections, and over 1.6 million mortalities occur each year because of serious fungal diseases [1,2]. Most fungal pathogens are opportunistic; they mainly infect immuno-compromised patients, such as patients with HIV, cancers, organ transplants, burns, diabetes mellitus, neutropenia, or chronic immunosuppression, patients taking broad-spectrum antibiotics, or patients in intensive care units. Fungal pathogens can also infect plants and foods, causing major losses in agricultural activities and food production [3]. Considering the current research and development (R&D) needs and perceived public health importance of this area, the World Health Organization (WHO) published the first fungal priority pathogens list, the WHO FPPL, in late 2022 [4], highlighting the urgent need for the development of novel antifungal drugs.

As an essential membrane-embedded enzyme in fungi, the plasma membrane H^+^-ATPase (Pma1) pumps protons out of the cell to maintain the transmembrane electrochemical gradient and membrane potential [5]. The proton gradient generated by this enzyme provides the energy for the active transport of nutrients. The intracellular pH of yeast is tightly regulated and physiologically maintained around neutrality [6]. The activity of Pma1 has been reported to be inhibited at a neutral pH and activated at an acidic pH [7]. C-terminal was found to be involved in this inhibition [8,9,10,11]. The plasma membrane H^+^-ATPase is conserved in plants and fungi (AHA2 in plants) but is not present in mammals. Therefore, Pma1 has been proposed to be a promising target for new broad-spectrum antifungal drugs, and many inhibitors of Pma1 have been reported.

The plasma membrane H^+^-ATPase belongs to the P3A subfamily of P-type ATPases, which utilize ATP hydrolysis and phosphorylation to pump ions against the gradient through the cell membrane [12]. According to previous studies, P-type ATPases transport their substrate through a cyclic transition of E1–E1P–E2P–E2 states, known as the “Post–Albers” model. There are five main subclasses (P1 to P5) for P-type ATPases [13,14,15,16], each of which transports different substrates (cations or heavy metals for the P1, P2, and P3 ATPases; phospholipid molecules for the P4 ATPases [17]; polyamines or transmembrane helices for the P5 ATPases [18,19,20,21]). Like other P-type ATPases, Pma1 has a conserved architecture consisting of a transmembrane domain (TMD) and three large cytosolic domains: an actuator domain (A domain), a nucleotide-binding domain (N domain), and a phosphorylation domain (P domain) (Figure 1) [14]. Notably, the plasma membrane H^+^-ATPase is the only type of P-type ATPase that forms a hexamer; all other members function as a monomer or a heterodimeric complex with an additional regulatory subunit (Figure 1). The Pma1 hexamer has been reported in *S. cerevisiae* [22], *N. crassa* cells [23], and also in *Kluyveromyces lactis* [24]. The structural basis for hexameric Pma1 assembly and how it functions have long been unclear.

Two articles, published in 2021, reported for the first time the high-resolution structure of Pma1 hexamers in autoinhibited and activated states [25,26]. In this review, we described the structural features, the activation mechanism, the proton transport mechanism, and the drug discoveries related to Pma1.

## 2. A Brief History of Pma1 Structure Determination

The high-resolution structural determination of Pma1 took a long time (Figure 2). The gene encoding the yeast plasma membrane H^+^-ATPase was first cloned in 1986 [12]. Earlier structural studies of Pma1 mainly focused on the plasma membrane H^+^-ATPase of *N. crassa* (ncPma1). As early as 1994, G.A. Scarborough succeeded in crystalizing ncPma1 [27]. In 1995, Scarborough and Kühlbrandt resolved the 2D crystal structure of ncPma1 at approximately 10.3 Å resolution and revealed that the crystalized repeating unit consisted of six 100 kDa ATPase monomers [28]. In 1998, the Kühlbrandt group improved the resolution of the Pma1 2D crystal structure to ~8 Å and showed that each ncPma1 monomer contains 10 α-helices of the membrane structural domains as well as four major cytoplasmic structural domains [29]. In 2002, Kühlbrandt reported a structural model of the proton pump based on the crystal structure of the calcium ion pump in the rabbit muscle plasma membrane, revealing the possible pathways and regulatory mechanisms of protons across the membrane [23]. In the same year, using single-particle Cryo-EM, Henderson’s lab resolved the 3D structures of ncPma1 in the E1 state at 17 Å resolution and in the E1P ADP state at 17.5 Å resolution. The structures confirmed the hexameric architecture of Pma1 in solution and also revealed large conformational changes for Pma1 in different states [30]. In 2005, Boutry’s lab used Cryo-EM to show that the plant plasma membrane H^+^-ATPase forms a hexameric structure in complex with the 14-3-3 protein [31]. In 2007, Oecking’s lab reported a Cryo-EM structure of plant PMA2 in complex with 14-3-3 at a resolution of ~35 Å [32]. In 2007, Nissen’s lab reported the first high-resolution structure of the plasma membrane H^+^-ATPase (*A. thaliana* AHA2) at a resolution of 3.6 Å [33]. However, probably due to lacking the C-terminal regulatory domain, this structure was found to be in a monomer state. The structure was further refined using dynamic simulations to improve inaccurate details in 2017 [34]. In 2021, our lab and Bublitz’s lab reported the high-resolution structures of hexameric yeast Pma1 and ncPma1 in the autoinhibited state, respectively (PDB: 7NY1 from *N. crassa* at 3.26 Å, 7VH5 from *S. cerevisiae* at 3.20 Å) (Figure 3a–c) [25,26]. Furthermore, our lab reported an activated Pma1 hexamer at an acidic pH (PDB: 7VH6 from *S. cerevisiae* at 3.80 Å). These latest structures are starting to reveal the molecular mechanism of the plasma membrane H^+^-ATPase.

## 3. Overall Structure of Pma1

The structures of *S. cerevisiae* and *N. crassa* Pma1 are largely similar, except for some domain shifts in different states (Figure 3a–c). Here, we mainly focus our discussion on the *S. cerevisiae* Pma1 structures, which have been determined in both the autoinhibited and activated E2P states.

In all the Cryo-EM structures, Pma1 maintained the hexamer architecture. Each Pma1 subunit contains conserved P-type ATPase domains: a 10 transmembrane helix containing TMD; the cytosolic P and N domains that are both inserted between TMH4 and TMH5; and the cytosolic A domain inserted between TMH2 and TMH3. The N-terminal peptide of Pma1 is largely disordered. The C-terminal autoinhibitory domain (880–918 in yeast Pma1, termed C-tail) forms two helices in the autoinhibited Pma1. Assembly of the Pma1 hexamer is mainly mediated by three interfaces: interactions between the TM domains of adjacent Pma1 subunits, ordered lipids within the hexamer ring, and the cytoplasmic C-tail (Figure 4a).

In each Pma1 subunit, TMH3, -5, and -7 line the interior surface of the hexamer ring, and TMH1, -2, -6, and -9 face the outside surface. TMH7 and TMH10 of one Pma1 subunit are closely stacked with TMH3 and TMH4 of the adjacent Pma1 subunit, forming a “V” shape. In fact, the interface is mainly mediated by the cytoplasmic ends of these TMHs. Specifically, several polar residues in TMH7/10 (Thr775, Thr776, Lys781, Gln786, and Arg857) interact with the Tyr314 of TMH3′ and Thr316 of TMH4′. Furthermore, this domain–domain interface is maintained in both self-inhibited and activated Pma1 structures [25,26].

In the previously low-resolution Cryo-EM structure of H^+^-ATPase [30,32], a bulk of densities were found in the center ring of the Pma1 hexamer. These densities were first identified to be 115 well-defined fatty acid acyl chains in the yeast Pma1 structure at high resolution (Figure 4b). Because no lipid was added during the purification process and the well-resolved lipid densities in Cryo-EM maps were very ordered, it indicated the lipids assemble with Pma1 endogenously. The structures showed each Pma1 subunit interacts extensively with lipids, and the interactions clearly stabilize the Pma1 hexameric architecture. Similar but weaker densities were also found in the high-resolution structure of ncPma1. The lipids are proposed to be a mixture of phospholipids and sphingolipids because the membrane compartment containing Pma1 was shown to be rich in sphingolipids. However, exact composition and physiological function of the lipids remain exclusive and await further study.

In the autoinhibited Pma1, the C-tail is ordered and mediates interactions with the two neighboring P domains (Figure 4c). Specifically, Ser911, His914, and Glu917 of the C-tail interact with Lys566, Asn577, Tyr579, and Arg583 of the P domain, while Arg898, Glu901, and Asp902 of the C-tail interact with Ser595′, Asp599′, and Arg625′ of the neighboring P domain [26]. Importantly, the C-tail was found to be disordered in the activated E2P state, indicating its important role in the activation of Pma1.

Interestingly, the modeling of the C-tail in the structures of *S. cerevisiae* and *N. crassa* Pma1 has not been in agreement (Figure 4d,e). Although corresponding densities of the C-tail in these two structures are in a similar position, they were assigned to different subunits of the Pma1 hexamer. In the structure of *S. cerevisiae* Pma1, the linker density connecting the C-tail and the TMH10 is continuous and of good quality despite the relatively lower resolution, so assignment of the C-tail to the Pma1 subunit is unambiguous in the structure of *S. cerevisiae* Pma1. In contrast, the density of the linker region in the structure of *N. crassa* Pma1 is barely visible and is thus unable to completely support the current assignment. Determining whether the C-tail of *N. crassa* Pma1 is different from that of *S. cerevisiae* Pma1 requires further study.

## 4. Autoinhibition and Activation of Pma1

The activity of Pma1 was autoinhibited at a neutral pH and activated at an acidic pH [7]. The C-tail has been proposed to be involved in the autoinhibition and activation of Pma1, since the plasma membrane H^+^-ATPase is activated when the hydrophilic C-terminus is removed in the gene or cleaved by a protease [11,35,36]. In *S. cerevisiae*, the addition of glucose to the growth medium leads to a rapid increase in H^+^ secretion due to the activation of the plasma membrane H^+^-ATPase [37]. The phosphorylation of the inhibitory C-tail mediates the glucose-dependent activation of Pma1 [38]. In plants, blue light activates stomatal defense cells through phosphorylation of the inhibitory C-tail of the plasma membrane H^+^-ATPase [39]. Several phosphorylation sites in the C-tail have been identified, including Ser899, Ser911, and Thr912 in *S. cerevisiae* [38,40,41]. In response to glucose stimulation, Thr912 was phosphorylated by protein kinase C1 [42,43], and Ser899 by another protein kinase, Ptk2 [44]. In contrast, the kinase phosphorylating Ser911 is still unknown. The phosphatase Glc7 can dephosphorylate Ser899 but not Ser911/Thr912 [41].

Recent structural studies of Pma1 have shown how an acidic pH and phosphorylation of the C-tail activate Pma1 (Figure 4c) [25,26]. As mentioned above, the C-tail of Pma1 in the autoinhibited state mediates the interaction between two neighbor P domains, including two intermolecular salt bridges (Arg898:Asp599′ and Glu901:Arg625′) and an intramolecular hydrogen bond (Ser911:Lys566) with the P domains. Due to the effect of pH on the ionization state of acidic and basic amino acids, the two salt bridges likely break at a lower pH, releasing the C-tail and leading to Pma1 activation [26]. Phosphorylation of Ser899, Ser911, and Thr912 may have a similar effect to that of an acidic pH, where phosphorylated Ser911 prevents the C-tail from binding to the P domain and phosphorylated Ser899 or Thr912 break the two salt bridges. Furthermore, the structure of Pma1 in the active state revealed that six C-tails released from the P domains moved to the center of the Pma1 hexamer and interacted with each other. In vitro purification of the SUMO-tagged C-tail (C-terminal 22 residues) supported the finding that the C-tail has the nature of oligomerization. C-tail oligomerization can stabilize the activated Pma1 in the hexamer form and also prevent its binding to the P domains.

Although no regulatory protein has been implicated in the activation of Pma1, studies have shown that activation of the plant plasma membrane H^+^-ATPase, AHA2, requires the 14-3-3 protein [45,46,47]. The 14-3-3 protein binds to the phosphorylated C-tail of AHA2, in which Thr947 is modified by protein kinase [48,49]. Single-particle Cryo-EM analysis showed that the complex of 14-3-3 with the AHA2 homologue PMA2 has six-fold symmetry, and the stoichiometry of 14-3-3 and PMA2 is 6:6 [31].

## 5. Proton Transport Channel of Pma1

Structural comparison between Pma1 in the autoinhibited and activated states showed that the TMH3–10 of all Pma1 subunits and lipids in the central ring keep still while TMH1–2 move with respect to the remaining TMD region (Figure 5a,b). This indicates that the putative proton transport channel is between TMH1–2 and TMH3–10. The putative proton transport path can be divided into three groups of polar/charged residues: the top (extracellular) proton exit site (D140, D143, R324, and D720); the middle proton transport path in the membrane bilayer (R695 and D730); and the bottom (cytosolic) proton entry site (Q125, N154, Q161, and E162). Among these residues, D730, as the only acidic residue in the middle of the membrane, likely plays a central role in proton transport. Notably, the equivalent residue of D730 in AHA2 (D684) was shown to be essential for proton transport [50,51]. Pma1 mutations of D730 or R695 have been shown to reduce ATPase activity [52]. Furthermore, the three charged groups mentioned above are separated by two groups of hydrophobic residues: the upper group near the exoplasmic side contains V146, L150, I331, and V723, and the lower group near the cytosolic side contains V336.

In the autoinhibited state, the middle-charged group is connected to the cytosolic proton entry point and is separated from the exoplasmic proton exit site by the upper hydrophobic group. This architecture allows the D730 in the middle of the lipid bilayer to be protonated. In the active state, TMH1–2 move downward by ~6.7 Å and rotate by ~40° from their position in the autoinhibited state. As a result, the middle-charged group is connected to the exoplasmic proton exit site and separated from the cytosolic proton entry point. Importantly, D730 in this architecture forms a salt bridge with R695 and is thereby deprotonated. The released proton is likely to be stabilized by D143 of the exoplasmic proton exit site, which forms a salt bridge with R324 in the autoinhibited state but moves toward Asp730 to accept protons in the active state (Figure 5a,b and Figure 6).

## 6. Transport Cycle of Pma1

According to previous structural studies of P-type ATPases, the catalytic cycle of a P-type ATPase includes the transition of the E1–E1P–E2P–E2 states [53,54,55,56,57]. The A, N, and P domains move dramatically in the cycle and drive the conformational change in the TMD required for substrate transport. The cycle requires binding of ATP in the N domain, autophosphorylation, and dephosphorylation of an Asp in the P domain. As discussed above, the Pma1 C-tail inhibits Pma1 activity by interacting with two neighboring P domains. Based on the structure of inhibited *S. cerevisiae* Pma1 and *N. crassa* Pma1 in complex with ADP, ADP/ATP is still able to bind the N domain of inhibited Pma1 and induce the movement of the A domain and N domain. However, locking of the P domain by the C-tail restricts the conformational change in TMD and thus inhibits Pma1 activity. Therefore, Pma1 only has basal activity in the autoinhibited state (Figure 6). In the activated E2P state structure of Pma1, the C-tail becomes disordered, releasing the P domain to participate in the catalytic cycle of Pma1. Consistent with this, activated Pma1 tightly couples ATP hydrolysis to proton transport, while autoinhibited Pma1 only partially couples it [58,59]. Whether the inhibited Pma1 can complete the cycle of the E1–E1P–E2P–E2 states is unknown.

Two activated plasma membrane H^+^-ATPase structures have been reported thus far: the C-terminal truncated plant AHA2 in the E1-ATP state and the *S. cerevisiae* Pma1 in the active E2P state [25,26,33]. Combining these two structures and homology models of Pma1 based on other published P-type ATPase structures, we proposed a complete proton transport cycle of Pma1 (Figure 6). Firstly, Pma1 is activated from the autoinhibited state to the E1 state by relocation of the C-tail. D730 in the middle of the lipid bilayer is accessible from the cytosol and is protonated in a low pH environment. Then, ATP binding induces the movement of the N and A domains toward each other, while the P domain stays still (E1-ATP state). Accompanied with the movement of the A domain, TMH1–2 move downward and rotate slightly, and D730 is still accessible from the cytosol in this state. Subsequently, the D378 in the P domain of Pma1 is phosphorylated through the hydrolysis of ATP into ADP (E1P state). After releasing the ADP from the N domain, the conserved TGES motif in the A domain of Pma1 approaches the phosphorylation of D378 and catalyzes the dephosphorylation (E2P state). The rearrangement of the A domain and P domain from the E1P to E2P states also induces dramatic movement and rotation of TMH1–2. As a result, the D730 in the middle of the lipid bilayer is deprotonated and stabilized by a salt bridge with R695. The released proton is likely to be transferred to the D143 of the exoplasmic proton exit site. Then, the phosphate releases from the P domain and couples with the shifting back of TMH1–2 (E2). As in the inhibited state, D143 forms a salt bridge with R324 and is deprotonated. Finally, the proton diffuses to the exoplasmic side and the Pma1 changes to the E1 conformation for another reaction cycle.

In the above catalytic transport model, conformational change in TMD occurs only in TMH1–2, which are at the outer periphery of the Pma1 hexamer. As a result, the proton transport movements in the TMD of one Pma1 subunit appear to not affect those in the neighboring subunits. If this is the case, how does the Pma1 hexamer work in the transport cycle? In our previous work, a cooperative all-or-none activation mechanism for the Pma1 hexamer was proposed based on the Cryo-EM structures [26]. In this model, activation of one Pma1 subunit in the hexameric ring will lead to the activation of all six Pma1 subunits at least in two ways. Firstly, activation of one Pma1 subunit involves the release of the C-tail. Because the C-tail stabilizes the two neighboring P domains, its release will also activate the corresponding neighbor Pma1 subunit. Secondly, activation of one Pma1 subunit involves a ~30 Å move of the P domain to the active position. The moved P domain will push on the neighboring P domain because of steric conflict if the remaining five subunits are in the autoinhibited state. Therefore, compared to the monomer state, the Pma1 hexamer can pump protons more rapidly, which may enable the yeast to respond to environmental stresses more rapidly and ensure rapid fermentative growth [6,60].

## 7. Drug Discovery Related to Pma1

As the P3A ATPase uniquely present in plants and fungi controls electrochemical gradients and cell membrane potentials [61], the plasma membrane H^+^-ATPase is essential for cell survival. Indeed, Pma1 deficiency in *S. cerevisiae* is lethal [12,62]. The sequence of Pma1 is highly conserved among different pathogenic fungi with an identity of about 50–96%. The primary sequences of plasma membrane H^+^-ATPases in yeast and plants also share high identity (eg. *Saccharomyces cerevisiae* Pma1 vs. *Arabidopsis thaliana* AHA2 at ~40%). In contrast, the sequence homology of Pma1 with P-type ATPases in mammals is generally less than 30%. Therefore, Pma1 is a promising antifungal drug target, and selective inhibitors of Pma1 have potential as broad-spectrum antifungal drugs. In contrast, the development of anti-plant pathogenic fungal pesticides targeting Pma1 needs to consider side effects because of the similarity between fungal Pma1 and plant AHA2.

Many Pma1 inhibitors have been reported, including chemically synthesized compounds such as Omeprazole [63,64], D-Peptides [65], ebselen and its analogs [66,67], and natural products such as chebulagic acid [68], tellimagrandin II [69], and Dichamanetin [70]. Studies have shown that antimalarial spiroindolone KAE609 (Cipargamin), hitachimycin, and NSC11668 are novel antifungal agents targeting Pma1 [71,72,73]. In an extensive randomized screening effort, 14 potential Pma1 inhibitors were identified with IC50 values ranging from 0.04 to 22.60 μM [74]. A major challenge in the development of Pma1 targeting inhibitors is to avoid targeting other P-type ATPases. Tetrahydrocarbazoles [75], o-hydroxybenzylated flavanones and chalcones [70], and pyrido-thieno-pyrimidines [74] have been proposed to potentially inhibit specifically Pma1.

Structure-based drug design (SBDD) can greatly accelerate the pace of drug discovery [76]. This method needs to obtain the structure of the drug target at suitable resolution and select a potential binding cavity of the structure first. Then, compounds or fragments of compounds from the huge databases of small molecules are docked into the cavity and scored using computer algorithms. The top hits are tested with in vitro activity assays to identify initial promising ligands. Subsequently, structure of the drug target in complex with these ligands can be determined, and the structural insights into the ligand–protein complex are used to further optimize the ligand. Usually, multiple iterations of ligand virtual screening, evaluation, and optimization are performed to increase the specificity and efficacy of the ligand. Finally, the best lead compounds can be evaluated in clinical trials.

Therefore, structural studies of Pma1 can help with the design and development of antifungal drug molecules targeting Pma1, facilitating the rapid, cost-effective, and efficient discovery of related lead compounds. Next question is which part of the Pma1 structure can be used as a potential ligand binding site for virtual screening. Several inhibitors of human P-type ATPases have been applied clinically, including cardiac glycosides targeting Na^+^/K^+^-ATPase and proton pump inhibitors (PPIs) targeting gastric H^+^/K^+^-ATPase. Both cardiac glycosides and PPIs inhibit P-type ATPases by blocking the extracellular substrate transporting channel between TMH1/2 and TMH4/6 [77,78]. However, the substrate transporting channel in Pma1 is much smaller and unfavorable for accommodating putative inhibitors. In contrast, two pockets (P1 and P2) adjacent to the substrate transporting channel of Pma1 seem to be more promising for the development of antifungal inhibitors (Figure 7a–c). These two pockets are both composed of TMH1–4 and are located at the extracellular leaflet and the intracellular leaflet of the membrane, respectively. Clearly, binding of inhibitors in either pocket will block the movement of TMH1–2. The Pma1 structures imply that TMH1–2 move dramatically during substrate transport cycle, and inhibitors in these two pockets likely block the activity of Pma1 and thus have potential as antifungal drugs. Notably, the proposed cipargamin binding site in PfATP4 of *P. falciparum* is located in a similar position to the second pocket in Pma1 [79].

## 8. Perspectives

Recent Cryo-EM structures of fungal H^+^-ATPases have provided insight into its mechanisms [5]. The hexameric architecture, ordered lipid disc in the hexamer center, location of the inhibitory C-tail, and dramatic conformational change in TMH1–2 provide a reliable working model for the activation and proton transport of Pma1. However, many questions remain to be answered. For example, the structures of activated Pma1 in the E1 and E2 states during the Post–Albers cycle are still unclear. How the six subunits in the Pma1 hexamer cooperate with each other remains elusive. Additionally, how the Pma1-specific inhibitors work is unknown. We look forward to finding the answers to these questions in the coming years.

## Figures and Tables

**Figure 1 jof-10-00273-f001:**
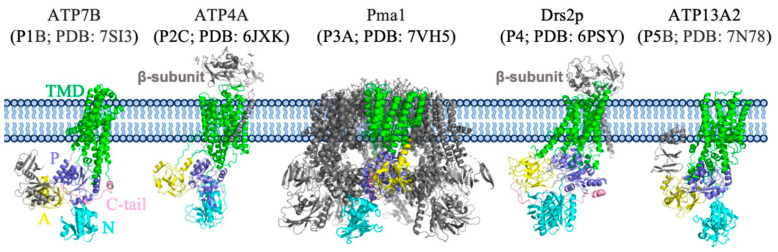
Structures of typical P-type ATPases in the P1–5 subfamilies. The major domains and subunits are labeled in different colors.

**Figure 2 jof-10-00273-f002:**
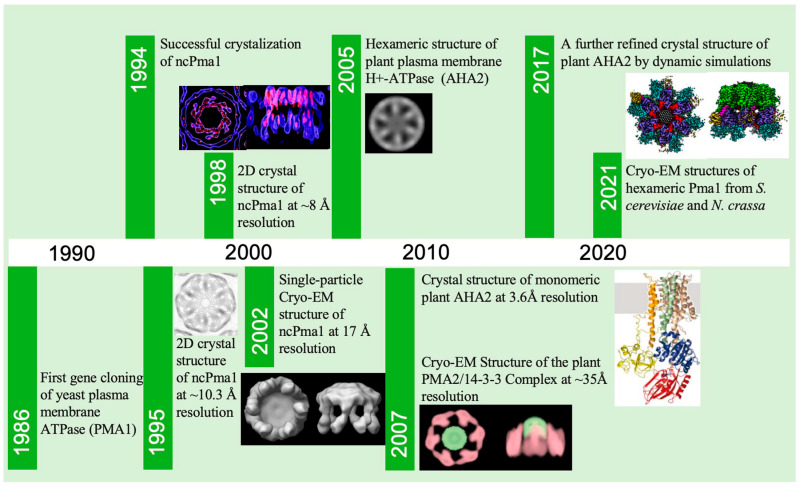
A timeline of landmark moments during plasma membrane H^+^-ATPase structure determination.

**Figure 3 jof-10-00273-f003:**
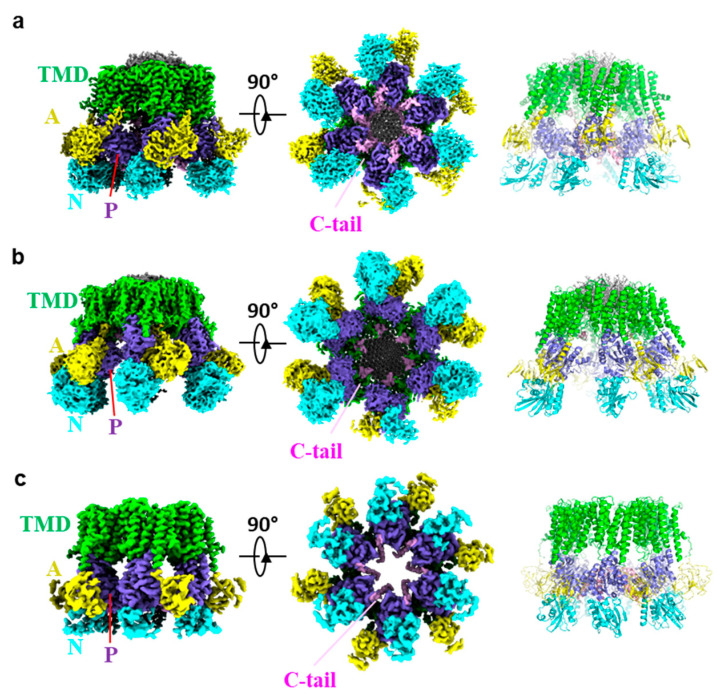
Cryo-EM structures of autoinhibited ((**a**), PDB ID: 7VH5), and activated ((**b**), PDB ID: 7VH6) *S. cerevisiae* Pma1, and autoinhibited *N. crassa* Pma1 ((**c**), PDB ID: 7NY1). The major domains are labeled in different colors.

**Figure 4 jof-10-00273-f004:**
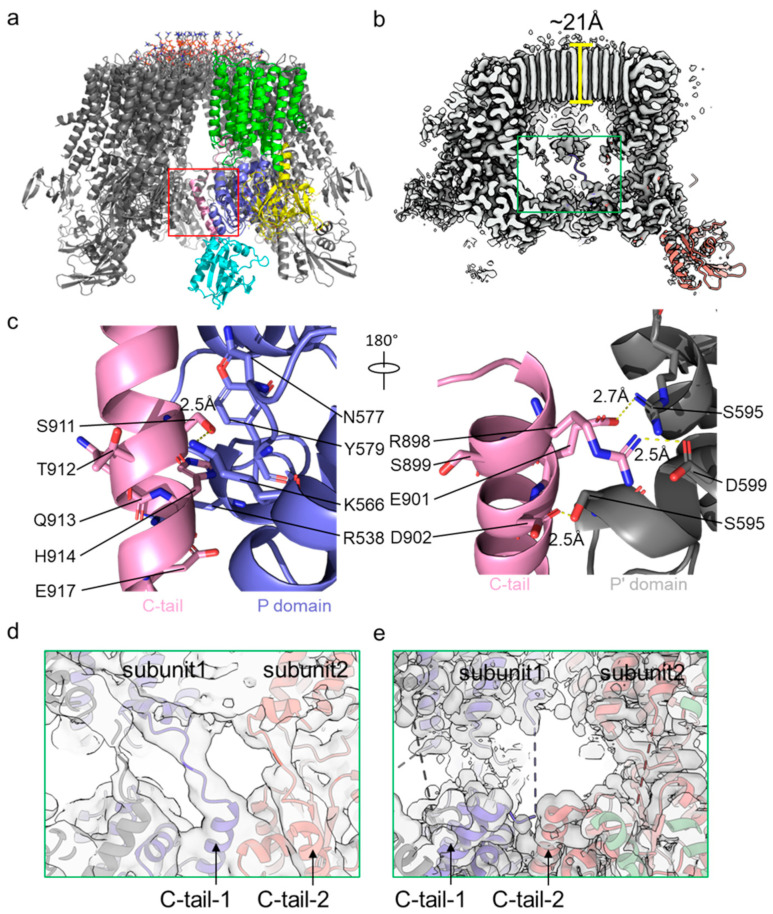
Mechanism of C-tail-mediated hexamer assembly and autoinhibition of Pma1. (**a**) Structure of autoinhibited Pma1. Major domains of one subunit are shown in multiple colors, while other subunits are in gray. The interface mediated by the C-tail is highlighted by a red rectangle. (**b**) Cut-in view of a Cryo-EM map of autoinhibited Pma1. The linker peptide region between the C-tail and TMH10 is highlighted by a green rectangle. (**c**) Detailed interactions of the C-tail with two neighboring P domains of Pma1. (**d**) The linker peptide density between the C-tail and TMH10 is shown at a lower threshold in the autoinhibited *S. cerevisiae* Pma1 (PDB ID: 7VH5). (**e**) The linker peptide density in the autoinhibited *N. crassa* Pma1 (PDB ID: 7NY1) is shown in same view as (**d**). The C-tail is assigned to different Pma1 subunits.

**Figure 5 jof-10-00273-f005:**
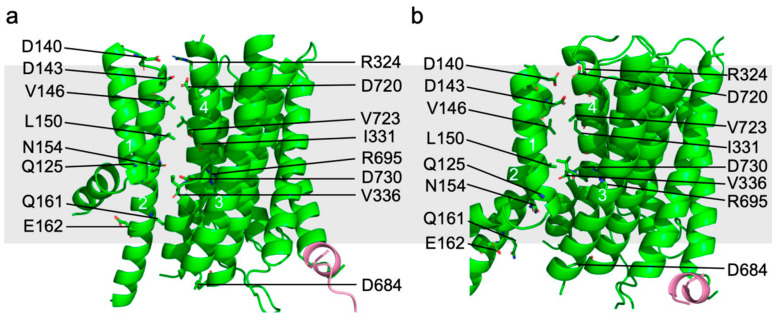
Proton transport channel of Pma1. Putative substrate transport path of Pma1 in the autoinhibited (**a**) and active state (**b**). TMH1–2 move downward by 6.7 Å and rotate by 40°. Residues lining the path are labeled in the figure and shown as sticks.

**Figure 6 jof-10-00273-f006:**
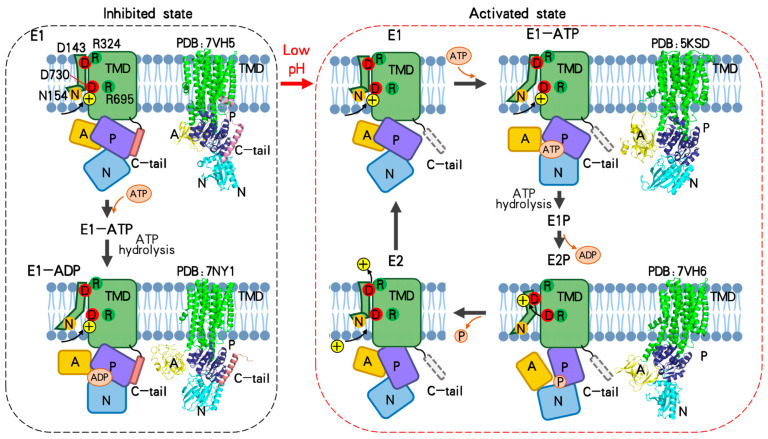
Model for proton transport across the plasma membrane by Pma1. According to the structure of inhibited *S. cerevisiae* Pma1 and *N. crassa* Pma1 in complex with ADP, ADP/ATP is able to bind the N domain of inhibited Pma1 and induce the movement of the A domain and N domain (**left panel**). However, the P domain is locked by the C-tail and thus Pma1 only has basal activity in the autoinhibited state (**left panel**). Pma1 is activated in acidic pH by releasing the C-tail from the P domain (**right panel**; released C-tail is shown in dotted box). In the proposed Post–Albers cycle of Pma1, D730 in the middle of the lipid bilayer is protonated in the E1 state and deprotonated in the E2P state. The released proton is transferred to D143 and then diffused to the extracellular side. Accompanied by the movement of TMH1–2 in different states, two salt bridges (D730:R695 and D143:R324) form alternately to transfer the proton. The structures of Pma1 in the E1, E1P, and E2 states are not yet resolved. More details are in the text.

**Figure 7 jof-10-00273-f007:**
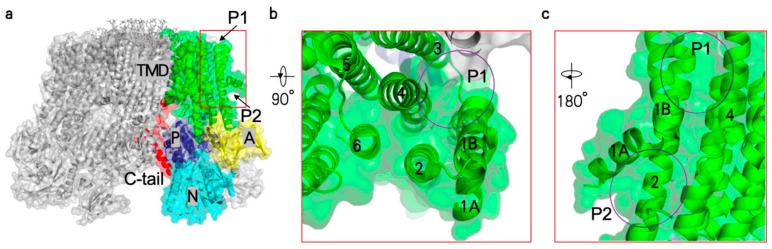
The two putative inhibitor binding pockets in Pma1. Pocket 1 (P1) and pocket 2 (P2), located at the extracellular leaflet and the intracellular leaflet of membrane, respectively (**a**). The major domains of Pma1 are labeled in different colors. Zoom-in views of the pockets are shown in (**b**,**c**). The positions of the pockets are highlighted by purple ovals. Numbers represent the serial numbers of the transmembrane helix.

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
