# Peer review of "Fungal Plasma Membrane H+-ATPase: Structure, Mechanism, and Drug Discovery"

_jof, 2024, doi:10.3390/jof10040273_

Round 1
Reviewer 1 Report
The paper by Chao-Ran and coworkers deals with the fungal proton ATPase Pma1, essentially from a mechanistic and structural point of view. This is an important component of fungal (and plant) biology and deserves attention. The work is quite sound from the scientific point of view. However, there are quite a few problems with the writing and the use of references, thus preventing acceptation in its present form. I’ll try to summarize the most relevant ones.
1.- Problems with references.
- Lines 51-60. Citations do not follow the correct increasing numbering and are not correctly attributed.
- Reference 6 and 53 are the same.
- Very important, Young and coworkers published a review exactly on the same topic several months ago (doi: 10.1016/j.bbamcr.2023.119600). While this do not detract merit from the present work, this reference needs to be included.
2. Writing:
Even not being myself an English native speaker, I noticed many places where writing can be improved. Examples are:
L. 17.- reviewed (change to “reviews”)
L. 51.- “starved or stressed yeast cells”, please use S. cerevisiae instead of “yeast”, as K. lactis and N. crassa are also yeasts.
L. 73.- plasma (membrane).
L. 80.- in complexed (in complex)
L. 102.- if not specifically mentioned (otherwise?)
L. 114 (legend). Major domains of one subunit(s) (use singular here)
L. 134. Endogenous (ly?)
L. 151. Whether the C-tail.. (Shouldn’t be “To elucidate whether…”?)
L. 164. “In contrast, the phosphorylated kinase of Ser911..” I feel the sentence should read “In contrast, the kinase phosphorylating Ser911..”
L. 264 “was found to be leath” (lethal?)
Other points
- please, use super index to denote the charge of the proton.
- L. 163.- “Thr912 was phosphorylated by protein kinase C1 [34] [35], and Ser899 by protein tyrosine kinase 2 (Ptk2)”. Please, note that Ptk2 is not a protein tyrosine kinase, but a Ser/Thr kinase. In this case, “Pt” probably derives from “polyamine transport” a phenotype found in the gene mutants. The alias for the gene (STK2) also points to this feature (Spermidine transport kinase 2).
- Figure 3. Colors used for “A” (yellow) and C-tail labeling are very hard to distinguish from the background. Please, improve this (maybe using dark background for the letters?)
L. 297. Please, indicate that PfATP4 corresponds to P. falciparum.
Reviewer 2 Report
The aim of the authors has been to review the current knowledge on the structure and mechanism of the fungal P-type ATPases and how this knowledge might be used to develop new antifungal drugs for Human use.
Globally, there are some references missing such as the WHO report on fungal infections or the paper describing the first ATPase sequence.
I think the authors should be clearer on the role of ATPase oligomerization, it is only said in passing that the C-tail of one inhibits the activity of a neighboring transporter.
Sometimes there is not enough information such as in the first paragraph where it is not mentionned that fungi are opportunistic pathogens, hence they infect weakened patients. That this research is aimed at treating mammalians since fungal anf plant P-type ATPases seem to share common regulatory mechanisms and that fungal infection are a huge agricultural problem.
I'm confused because lines 50-52 the authors specify that the hexameric state of the H+ ATPase is found in starving cells of several fungi and for example line 102 they seem to treat it as the normal state of the ATPase. And nowhere do they show if this oligomerization state is the one found in cells.
line 39 I think it is worth mentionning that the transporter (it is not enzyme as stated in line 31) does it against the proton gradient and more generally that their main function transporting against the gradient.
line 133-134, the authors justify the validity of the hexamer state and the presence of lipids at the center of the ring because no lipids were added during purification. This could be the result of the extraction that requires the hydrophobic regions of the ATPase to be protected from the aquous solvent. I think that better argument would be that if this were the case, no structure would be resolved because of the aphazardous organization of the proteins. The regularity of the purified structure validates this arrangement.
Finally, there are some improvements to be made on the writing. Sentences 13,198 for example are unclear. There are also typossuch as line 28 (an not required), lines 122-123 (three TMHs not required, cite them directly) or line 264 leathal?
Round 2
Reviewer 2 Report
The most important comment is that the authors have to develop the part about the methods for drug discovery based on structural knowledge. Are they envisaging docking, co-crystalization or other means to providre insights in drug discovery?
I have issues with the model proposed in figure 6. It is based on structures but the transition from one to the other is sometimes hard to follow. For example , where does the ADP come from in the inhibited state, does a drop in intracellular pH induce the H+-ATPase from inhibited to actived state? The model heavly relies on activated structures that do not have a C-terminus if I go by the dotted lines, if so, it is not in the legend of the figure.
The second comment is about the aim of drug discovery. Indeed, we have to surmise that the aim is medical and not agricultural since Plant and Fungi have highly similar H+-ATPase compared to animal. So drug discovery against fungi would also affect plants. This is relevant because fungi also are detrimental in agrosystems.
line 37 ...provides THE energy for...
line 294 Studies have shown that the antimalarial spiroindolone KAE609 (), hitachimycin...
Author Response
Reviewer2#
1.The most important comment is that the authors have to develop the part about the methods for drug discovery based on structural knowledge. Are they envisaging docking, co-crystalization or other means to providre insights in drug discovery?
Answer response: We are grateful to the reviewer for the valuable comment. A paragraph was added to introducing the process of SBDD and explain how the structural knowledge can be used in the process:
“Structure-based drug design (SBDD) can greatly accelerate the pace of drug discovery. This method needs to get the structure of drug target and select a potential binding cavity of the structure first. Then, compounds or fragments of compounds from the huge databases of small molecules are docked into the cavity and scored using computer algorithms. The top hits are tested with in vitro activity assays to get initial promising ligands. Subsequently, structure of the drug target in complex with these ligands can be determined, and the structural insights into the ligand–protein complex can be used to further optimize the ligand. Usually, multiple iterations of ligand virtual screening, evaluation and optimization are performed to increase the specificity and efficacy of the ligand. Finally, the best lead compounds can be evaluated in clinical trials.
Therefore, structural studies of Pma1 can help with the design and development of antifungal drug molecules targeting Pma1, facilitating the rapid, cost-effective, and efficient discovery of related lead compounds. Next question is which part of the Pma1 structure can be used as potential ligand binding cactiy for virtual screening. ”
2.I have issues with the model proposed in figure 6. It is based on structures but the transition from one to the other is sometimes hard to follow. For example , where does the ADP come from in the inhibited state, does a drop in intracellular pH induce the H+-ATPase from inhibited to actived state? The model heavly relies on activated structures that do not have a C-terminus if I go by the dotted lines, if so, it is not in the legend of the figure.
Answer response:We apologize for the unclear figure and are grateful to the reviewer for the valuable comment. We have revised the legend and the figure.
Firstly, in bottom part of the left panel, the structure of inhibited N. crassa Pma1 in complex with ADP was present. The ADP was incubated with Pma1 for structure determination, but should be generated from ATP hydrolysis in real substrate transport cycle. Therefore, we have added the ATP binding and ATP hydrolysis in the figure, and revised the legend to “According to the structure of inhibited S. cerevisiae Pma1 and N. crassa Pma1 in complex with ADP, ADP/ATP is able to bind the N domain of inhibited Pma1 and induces the movement of the A domain and N domain (left panel). However, the P domain is locked by the C-tail and thus Pma1 only has basal activity in the autoinhibited state (left panel). ”.
Secondly, the H+-ATPase is indeed activated by the drop of intracellular pH. The figure was revised by adding “low pH” above the transition arrow, and the legend was revised to “Pma1 is activated in low pH by releasing the C-tail from the P domain (right panel).”
Thirdly, the C-tail was discordered in activated S. cerevisiae Pma1 structure. Cryo-EM structures of inhibited S. cerevisiae and N. crassa Pma1 have revealed the C-tail inhibits Pma1 activity by interacting with two neighboring P domains via two important inter-subunit salt bridges. At low pH, the salt bridges may break and release the inhibitory C-tail, leading to the activation of Pma1. In fact, the released C-tail are likely to be oligomerized in the center of activated Pma1 hexamer. We have explained “released C-tail is shown in dotted box” in the legend.
3.The second comment is about the aim of drug discovery. Indeed, we have to surmise that the aim is medical and not agricultural since Plant and Fungi have highly similar H+-ATPase compared to animal. So drug discovery against fungi would also affect plants. This is relevant because fungi also are detrimental in agrosystems.
Answer response:We are grateful to the reviewer for the valuable comment. We have added related discussion:
“The sequence of Pma1 is highly conserved among different pathogenic fungi, with an identity of about 50–96%. Sequence of plasma membrane H+-ATPases in yeast and plant also share high identity (eg. Saccharomyces cerevisiae Pma1 vs Arabidopsis thaliana AHA2 at ~40%). In contrast, the sequence homology of Pma1 with P-type ATPases in mammals is generally less than 30%. Therefore, Pma1 is a promising antifungal drug target, and selective inhibitors of Pma1 have potential as broad-spectrum antifungal drugs. In meanwhile, the development of anti-plant pathogenic fungal pesticides targeting pma1 need to consider side effect because of the similarity between fungal Pma1 and plant AHA2.”
4.line 37 ...provides THE energy for...
Answer response:Thank you for the comment. The word was added.
5.line 294 Studies have shown that the antimalarial spiroindolone KAE609 (), hitachimy
Answer response:Thank you for the comment. The sentence was corrected.

Round 3
Reviewer 2 Report
The authors do not contexualize the findings. The ATPase functions on the plasma membrane of a cell. Lines 37-38 they report the findings of the Slayman lab but fail to mention that the result is obtained from Pma1 on secretory vesicles hence the pH tested is the intracellulat pH. They often talk about low pH without mention what it means physiologically. van Euwen et al (2010) measure the intracellualr pH at 6.8. How does that fit with their data.
They mention that the Pma1 hexamer contains lipids but do we know which one and if they are required for the proper function of the ATPase. If so, this could also be an interesting fungicide target.
lines 165-168, I think that physiologically, the addition of glucose results in the increase in H+ intracellularly which induces Pma1 activity.
Finally, if I understood figure 6, based on biochemical and structural evidence, Pma1 can hydrolyse ATP to generate ADP which binds Pma1 in one condition and in another Pma1 binds ATP to phosphorylate Pma1 which results in proton excretion. Is it correct to say that in stationary phase Pma1 binds ADP, cells have a low energy level and in high energy level, exponential growth, it binds ATP to excrete protons?
line 41 a promising target
line 242 ital cerevisiae
line 298 capitalize Pma1
